# The Prognostic Value of Optic Nerve Sheath Diameter/Eyeball Transverse Diameter Ratio in the Neurological Outcomes of Out-of-Hospital Cardiac Arrest Patients

**DOI:** 10.3390/medicina58091233

**Published:** 2022-09-06

**Authors:** Byeong-In Cho, Heekyung Lee, Hyungoo Shin, Changsun Kim, Hyuk-Joong Choi, Bo-Seoung Kang

**Affiliations:** Department of Emergency Medicine, Hanyang University College of Medicine, Seoul 04763, Korea

**Keywords:** cardiac arrest, optic nerve sheath diameter, eyeball transverse diameter, neurologic outcome, out-of-hospital cardiac arrest

## Abstract

*Background and objectives*: The optic nerve sheath diameter (ONSD) is indicative of elevated intracranial pressure. However, the usefulness of the ONSD for predicting neurologic outcomes in cardiac arrest survivals has been debatable. Reportedly, the ONSD/eyeball transverse diameter (ETD) ratio is a more reliable marker for identifying intracranial pressure than sole use of ONSD. *Materials and Methods*: This retrospective study aimed to investigate the prognostic value of the ONSD/ETD ratio in out-of-hospital cardiac arrest (OHCA) patients. We studied the brain computed tomography scans of adult OHCA patients with return of spontaneous circulation, who visited a single hospital connected with a Korean university between January 2015 and September 2020. We collected baseline characteristics and patient information from electronic medical records and ONSD and ETD were measured by two physicians with a pre-defined protocol. According to their neurologic outcome upon hospital discharge, patients were divided into good neurologic outcome (GNO; cerebral performance category [CPC] 1–2) and poor neurologic outcome (PNO; CPC 3–5) groups. We evaluated the ONSD/ETD ratio between the GNO and PNO groups to establish its prognostic value for neurologic outcomes. *Results*: Of the 100 included patients, 28 had GNO. Both the ONSD and ETD were not significantly different between the two groups (ONSD, 5.48 mm vs. 5.66 mm, *p* = 0.054; ETD, 22.98 mm vs. 22.61 mm, *p* = 0.204). However, the ONSD/ETD ratio was significantly higher in the PNO group in the univariate analysis (0.239 vs. 0.255, *p* = 0.014). The area under the receiver operating characteristic curve of ONSD/ETD ratio for predicting PNO was 0.66 (95% confidence interval, 0.56–0.75; *p* = 0.006). There was no independent relationship between the ONSD/ETD ratio and PNO in multivariate analysis (aOR = 0.000; *p* = 0.173). *Conclusions*: The ONSD/ETD ratio was more reliable than sole use of ONSD and might be used to predict neurologic outcomes in OHCA survivors.

## 1. Introduction

Ischemia-reperfusion cerebral injury after the return of spontaneous circulation (ROSC) from cardiac arrest (CA) may increase intracranial pressure (ICP) and cause cerebral edema [1]. Neuroimaging may be helpful in detecting and quantifying cerebral injury in post-CA syndrome (PCAS) [2]. The optic nerve sheath diameter (ONSD) can be measured using neuroimaging tools, and increased ONSD is associated with raised ICP [3]. Thus, ONSD measurement may be useful for predicting neurological outcomes in post-CA patients.

ONSD evaluated using brain computed tomography (CT), ultrasonography (US), and magnetic resonance imaging (MRI) may be useful for predicting neurological outcomes in PCAS [3,4,5]. However, a recently published meta-analysis demonstrated that ONSD measurement was not significantly effective in predicting survival outcome and neurologic outcome [6]. The sole use of ONSD may be affected by sex, body mass index (BMI), and head circumference in patients with CA. Lee et al., attempted to identify differences in ONSDs pre-CA and post-CA using brain CT imaging [7]. This study concluded that the amount and rate of ONSD changes, rather than ONSD itself, were related with neurological outcome in resuscitate patients from CA. However, ONSD measurements by both pre- and post-CA brain CT cannot be routinely performed.

In healthy people, a substantial association between ONSD and eyeball transverse diameter (ETD) has been established [8]. Thus, the ONSD/ETD ratio has potential use in the evaluation of patients with intracranial hypertension after anoxic brain injury [9]. Using brain CT imaging, this study tried to identify the relationship between the ONSD/ETD ratio and neurologic outcomes in OHCA survivors.

## 2. Materials and Methods

### 2.1. Study Design and Setting

Between January 2015 and September 2020, patients hospitalized with ROSC after OHCA at a single hospital connected with a Korean university were included in this retrospective observational cohort study. The Institutional Review Board (IRB) of Hanyang University Guri Hospital approved this research (IRB No. GURI 2020-12-008). Because of the retrospective nature of the investigation, informed consent was not required.

### 2.2. Participants

Adult patients who visited to the emergency department (ED) after OHCA and had undergone a brain CT scan after ROSC were included in this study. The exclusion criteria were as follows: (1) age under 19 years; (2) intracranial lesions, including supratentorial and infratentorial lesions, such as cerebral hemorrhage or brain tumor; (3) ophthalmologic problems that could influence the ONSD or eyeball; and (4) transfer to another hospital. According to neurologic outcome on hospital discharge, included patients were classified into two groups: good neurologic outcome (GNO) and poor neurologic outcome (PNO). Brain CT had been obtained in the ED and was prior to apply targeted temperature management (TTM) in all patients. Further, the ONSD/ETD ratio on brain CT was measured and analyzed between-group analysis. The relationship between the neurologic outcome and the ONSD/ETD ratio of patients resuscitated following OHCA was the primary outcome.

### 2.3. Data Collection

The following basic characteristics of included patients were collected retrospectively from the electronic health record: age, sex, comorbidities, etiology, witnessed arrest, bystander cardiopulmonary resuscitation (CPR), the initial cardiac rhythm (shockable vs. non-shockable), no-flow time (the time between recognize CA and start CPR) and low-flow time (the time between start CPR and ROSC), pre-hospital ROSC, and the time interval between ROSC and brain CT. We also used the cerebral performance categories (CPC) to obtain neurologic outcomes at hospital discharge. GNO and PNO were defined as CPC scores of 1–2 and 3–5, respectively.

### 2.4. ONSD/ETD Ratio Measurements

Brain CT were scanned parallel to the orbital floor from the base of the skull to the vertex using 4 mm non-contrast continuous slices, according to normal methods. ONSD and ETD are the distances between the outer margins of the thick sheath layers surrounding the optic nerve and the globe, respectively. The ONSDs were measured estimated at 3 mm behind the globe in both sides, and the axial scan image with the maximum ETD of each eye was chosen. The ONSDs and ETDs of both the eyes were averaged to obtain mean values, and the ONSD/ETD ratio was calculated. All measurements, including neurological outcomes, were carried out by two emergency physicians who were blinded to patient information. The average values of the ONSD and ETD measured by the two physicians were adopted. The images were enlarged by 450% using the PACS tool and moved to 440 of the window width and 45 of window level. The mean values of ONSDs and ETDs were obtained from the right and left eyes. Additionally, we calculated the ONSD/ETD ratio. SOMATOM Definition Edge, SOMATOM Definition DS, and SOMATOM Sensation 16 CT equipment were utilized (Siemens Healthcare, Erlangen, Germany). The following parameters were set: 120 kVp, 250–500 mAs, and a slice thickness of 4–4.5 mm. All CT scans were stored in the PACS by DICOM (digital imaging and communications in medicine) format.

### 2.5. Study Size

The study size was estimated based on a pilot study with 30 patients. Patients with GNO and PNO had ONSD/ETD ratios of 0.238 ± 0.026 and 0.253 ± 0.032. The needed sample size was determined to be 96 individuals (a-error = 0.05; power = 0.8; effect size = 0.514), and 107 participants were required to consider for a 10% dropout rate.

### 2.6. Statistical Methods

The number with percentages and median with interquartile range (IQR) were used to report continuous and categorical variables, respectively. The Shapiro–Wilk test was used to analyze non-normally distributed variables, whereas the Wilcoxon rank-sum test were used to analyze normally distributed variables. The chi-square test or Fisher’s exact test were used to analyze categorical variables. For analyses, differences with a *p*-value of <0.05 were considered statistically significant. The odds ratio of each covariant for PNO were determined using multivariate analysis with logistic regression, with adjustments for confounding variables that were shown to be significant in the univariate analysis. In the multivariable analysis, variables with *p* < 0.1 in univariate analysis with the ONSD/ETD ratio were included. The Hosmer–Lemeshow test was used to validate the calibrations of the logistic model. The area under the receiver operating characteristic (ROC) curve (AUC) was utilized to evaluate the prognostic usefulness of the ONSD/ETD ratio for predicting neurologic prognosis, with a sensitivity greater than 1 indicating specificity. The Youden index was used to obtain the results, which were then presented as a 95% confidence interval (CI) of AUC with sensitivity, specificity, positive predictive value (PPV), and negative predictive value (NPV). The intraclass correlation coefficient was calculated to evaluate inter-rater reliability for ONSD/ETD ratio measurement between two physicians. MedCalc Statistical Software (version 17.2, MedCalc Software, Ostend, Belgium) was utilized for ROC analysis, G*Power (3.1.9.6; Heinrich Heine University, Düsseldorf, Germany) was used for study size calculation, whereas SPSS software (version 25.0, IBM, Armonk, NY, USA) was used for other statistical study.

## 3. Results

### 3.1. Participants

In this study, 157 OHCA survivors who had brain CT following OHCA were enrolled over the research period. However, the following 57 patients were excluded: 2 patients who were <19 years of age, 11 patients with cerebral or subdural bleeding, 2 patients with a history of ophthalmologic condition or surgery, and 42 patients moved to another hospital. Further, 100 individuals were ultimately assigned to the GNO (n = 28, 28.0%) or PNO (n = 72, 72.0%) groups (Figure 1). The patients’ baseline characteristics are summarized in Table 1. The median age of the included patients was 66 (IQR: 56–77) years and 65.0% of them were male. Patients in the PNO group were much older than those in the GNO group, and the PNO group demonstrated a significantly greater incidence of diabetes mellitus and respiratory etiology. The PNO group had a lower proportion of bystander CPR, shockable rhythm, and longer no-flow and low-flow periods compared to the GNO group. In addition, the PNO group had a considerably lower pre-hospital ROSC and a longer time between ROSC and CT.

### 3.2. Comparison of the ONSD/ETD Ratio between GNO and PNO

The mean ONSD was 5.48 (5.21–5.87) and 5.66 (5.43–6.05) mm in the GNO and PNO groups (Table 2, Figure 2). There were no significant differences in ONSDs between the GNO and the PNO groups (*p* = 0.054). Moreover, there was no significant difference in ETDs across the groups (*p* = 0.204). However, the ONSD/ETD ratio was significantly higher in the PNO group (0.239 vs. 0.255, *p* = 0.014). The univariable and multivariable logistic regression for GNO with baseline variables and the ONSD/ETD ratio is summarized in Table 3. Age and low-flow time were independently associated with neurological outcomes. However, there was no independent association between the ONSD/ETD ratio and GNO in the multivariate analysis (adjusted odds ratio [aOR] = 0.000; *p* = 0.173). The excellent reliability was observed between the ONSD/ETD ratio measurements by two physicians (intraclass correlation coefficient; 0.951).

### 3.3. Diagnostic Value of the ONSD/ETD Ratio for Predicting the Neurologic Outcome

In the ROC curve for the ONSD/ETD ratio, the AUC for predicting PNO was 0.66 (95% CI, 0.56–0.75; *p* = 0.006; Figure 3). Patients with an ONSD/ETD ratio > 0.290 had PNO with 100% specificity and 100% PPV. GNO could be predicted with a cut-off of ≤0.257 using the Youden index in the ROC curve for the ONSD/ETD ratio, with 85.7% sensitivity and 45.8% specificity; the PPV and NPV were 38.1 and 89.2%, respectively (Table 4).

## 4. Discussion

In this study, we found that the ONSD/ETD ratio was significantly associated with neurological outcomes in patients with OHCA. However, neither ONSD nor ETD alone differed between the GNO and PNO groups. Nevertheless, there was no independent association between the ONSD/ETD ratio and neurological outcome after adjusting for confounders. Additionally, there was a slight difference in the ONSD/ETD ratio between the GNO and PNO groups and usefulness of cut-off value in clinical setting is limited because of the relatively small sample size.

A positive correlation between increased ICP and ONSD has already been reported in previous studies [10,11]. The invasive method for measuring increased ICP has the risk of causing severe complications, including intracranial hemorrhage, infection, and mechanical failure [12]. The ONSD measured on non-contrast brain CT showed a linear association with ICP obtained from invasive monitoring, and sonographic measurement would be useful for assessing increased ICP with a low risk of iatrogenic injury [11,13].

Several previous studies have reported that ONSD could be used as a predictor for neurologic outcomes in post-CA [7,14,15]. One retrospective study reported a close correlation between increased ONSD on brain CT and the neurologic outcome of hypoxic brain injury in post-CA patients [16]. A prospective pilot study from France reported ONSD as a promising tool for the early assessment of outcomes in post-CA patients [17]. Park et al., demonstrated that ONSD measured by US at 24-h after ROSC is a valuable tool to predict neurologic outcome in post-CA patients treated with TTM [4]. Furthermore, a meta-analysis reported that ONSD could be used as a predictor of neurologic outcomes in post-CA patients [14]. However, other studies have reported different results regarding the relationship between ONSD and neurological outcomes. One registry-based multicenter study from Korea reported that ONSD on initial brain CT was not correlated with neurologic outcomes at 6 months in post-CA patients treated with TTM [18]. Rush et al., reported that ONSD was not different between the GNO and PNO groups in post-CA patients [19]. Owning to the inconsistencies in previous study results regarding the predictive value of ONSD, a recent meta-analysis reported that it is difficult to determine the usefulness of enlarged ONSD in predicting neurologic outcomes post-CA [6].

Although ONSD could be considered an indirect marker for raised ICP, there are individual differences in ONSD in baseline healthy conditions [8,19,20]. The ONSD could be affected by individual characteristics, including age, race, or BMI [8,19]. Ultrasonographic measurements of healthy Asian adults revealed that a longer ONSD is correlated with male sex and a high BMI [8,19]. Owing to the limitations of using ONSD alone, some other studies used ONSD combined with other indicators, such as cerebrospinal fluid pressure, gray-to-white matter ratio, and serum albumin levels, to improve the predictive performance [3]. Lee et al., reported that ONSD changes between pre- and post-CA showed better predictive performance than sole use of ONSD to predict neurologic outcomes in survivors from CA [7]. However, there was a limitation for clinical use since a pre-CA brain CT is necessary to calculate ONSD changes.

The ONSD/ETD ratio was used to adjust the individual characteristics that could affect the ONSD. A study of healthy Korean adults found that only ETD was independently associated with ONSD in a multivariable analysis, and the ONSD/ETD ratio was not correlated with age, sex, BMI, mean arterial blood pressure, or intraocular pressure [21]. One retrospective study reported that the ONSD/ETD ratio was more accurate than ONSD as a predictor in comatose patients with supratentorial lesions [22]. Another study reported that the ONSD/ETD ratio might be a valuable marker for predicting increased ICP in traumatic brain injury [23]. However, Lee et al., reported that the ONSD/ETD ratio did not differ significantly between GNO and PNO patients having TTM after CA [18].

In this study, an increased ONSD/ETD ratio was more accurate than the sole use of ONSD or ETD in predicting PNO in OHCA survivors. This result may have been derived from the ONSD/ETD ratio, which was less affected by individual characteristics. However, the multivariable analysis failed to prove statistical significance in the present study. The performance of ONSD measured immediately after ROSC for prognostication of post-CA patients was limited. ICP elevation was reported to commence 24 h after ROSC, and the ONSD at 24 h after CA showed a higher sensitivity to predict neurologic outcome for post-CA patients in a previous study [4,24,25]. In another previous study on the relationship between ONSD, ICP, and neurologic outcomes in CA survivors, ONSD at 3 days after CA had an excellent correlation with ICP [5]. The median ROSC to CT interval was 97 min in this study, and the relatively low predictive performance of the ONSD/ETD ratio may have originated from this measurement time. Further study is required to reveal the optimal measurement time of ONSD/ETD ratio for a higher predictive performance.

There are some limitations in this study. First, this was a single-center study with a rather small study size, which could have resulted in insufficient statistical power; nonetheless, we calculated the study size and met the minimum requirement. Second, brain CT could not be scanned parallel to the optic nerve sheath, resulting in an oblique image of the optic nerve sheath or the eyeball rather than a horizontal image. Additionally, given the extremely small size of the ONSD on brain CT, there could have been slight measurement mistakes; however, to reduce these errors, two physicians who were blinded to patient information performed measurements using an established procedure with consensus. Third, despite our efforts to collect as many variables as feasible, there may be hidden confounding factors that could potentially affect the ICP, including the absence or presence of TTM, hyperosmolar therapy, positive pressure ventilation, or cardio-hemodynamics statement. Furthermore, the statistical difference in baseline characteristics between the GNO and the PNO groups may affect the results, and the individual difference of ROSC to CT time may have influenced the results. Fourth, recent guidelines propose neurologic outcome assessment 3 months following discharge [26]. However, we only examined the neurological outcome at the time of discharge and did not assess the long-term results. Fifth, this was a retrospective study, and the clinical relevance of its prognostic value remains unknown. A large-scale prospective study with serial measurement of the ONSD/ETD ratio using multiple modalities, including US or MRI, is needed to enhance our findings.

## 5. Conclusions

The ONSD/ETD ratio was more reliable than ONSD or ETD alone in predicting neurological outcomes in OHCA survivors and could have clinical application.

## Figures and Tables

**Figure 1 medicina-58-01233-f001:**
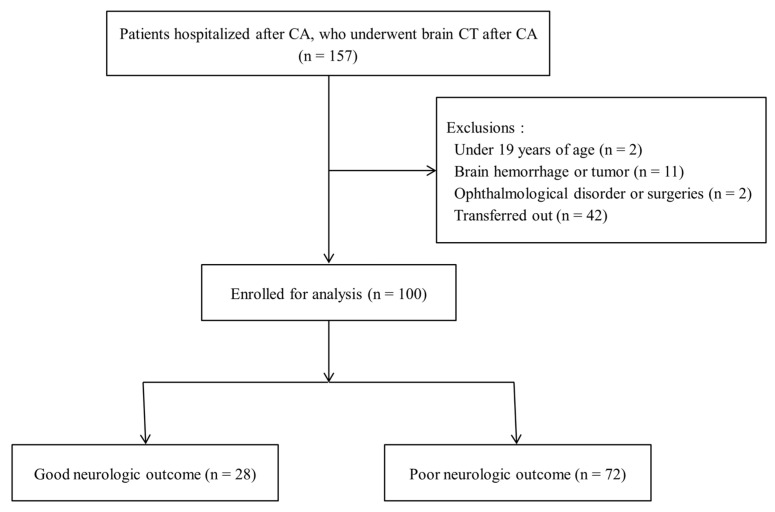
Flow chart of the study. Abbreviations: CA, cardiac arrest; CT, computed tomography.

**Figure 2 medicina-58-01233-f002:**
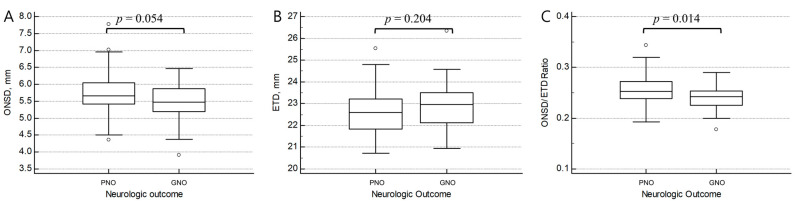
Comparisons of the optic nerve sheath diameter (**A**), eyeball transverse diameter (**B**), and ONSD/ETD ratio (**C**) between the good and poor neurologic outcome groups. Abbreviations: ETD, eyeball transverse diameter; GNO, good neurologic outcome; ONSD, optic nerve sheath diameter; PNO, poor neurologic outcome.

**Figure 3 medicina-58-01233-f003:**
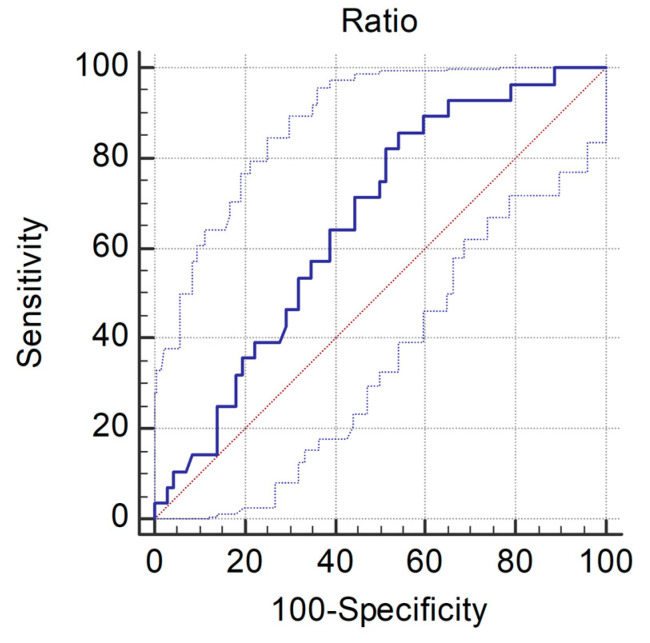
Receiver operator curve for predicting poor neurologic outcome using ONSD/ETD ratio. AUC = 0.66 (95% confidence interval = 0.56–0.75).

**Table 1 medicina-58-01233-t001:** Baseline characteristics and univariate analysis.

	Total (n = 100)	GNO (n = 28)	PNO (n = 72)	*p*-Value
Demographics				
Age, years	66 (56–77)	56 (52–65)	74 (60–79)	<0.001
Sex, male	65 (65.0)	22 (78.6)	43 (59.7)	0.076
Comorbidities				
HTN	49 (49.0)	14 (50.0)	35 (48.6)	0.901
DM	30 (30.0)	4 (14.3)	26 (36.1)	0.032
MI	12 (12.0)	2 (7.1)	10 (13.9)	0.501
Etiology				
Cardiac	31 (31.0)	22 (78.6)	9 (12.5)	<0.001
Respiratory	31 (31.0)	2 (7.1)	29 (40.3)	0.001
Others	38 (38.0)	4 (14.3)	34 (47.2)	0.002
Resuscitation				
Witnessed	66 (66.0)	19 (67.9)	47 (65.3)	0.807
Bystander CPR	62 (62.0)	22 (78.6)	40 (55.6)	0.033
Shockable rhythm	22 (22.0)	18 (64.3)	4 (5.6)	<0.001
Pre-hospital ROSC	47 (47.0)	23 (82.1)	24 (33.3)	<0.001
No-flow time, min	10 (1–21)	2 (0–9)	12 (5–26)	<0.001
Low-flow time, min	12 (8–23)	10 (6–16)	14 (9–25)	0.024
ROSC to CT interval *, min	97 (47–137)	58 (31–87)	113 (62–154)	<0.001
TTM	37 (37.0)	11 (39.3)	26 (36.1)	0.812

Abbreviations: CPR, cardiopulmonary resuscitation; CT, computed tomography; DM, diabetes mellitus; GNO, good neurologic outcome; HTN, hypertension; MI, myocardial infarction; PNO, poor neurologic outcome; ROSC, return of spontaneous circulation; TTM, targeted temperature management. * The interval between ROSC and brain CT.

**Table 2 medicina-58-01233-t002:** Comparison of the ONSD/ETD ratio between good and poor neurologic outcomes.

	Total (n = 100)	GNO (n = 28)	PNO (n = 72)	*p*-Value
Optic nerve sheath diameter, mm				
ONSD, Rt.	5.74 (5.31–6.15)	5.48 (5.19–5.96)	5.79 (5.43–6.19)	0.089
ONSD, Lt.	5.68 (5.22–5.90)	5.48 (5.16–5.79)	5.71 (5.24–6.06)	0.078
ONSD, average	5.64 (5.34–6.02)	5.48 (5.21–5.87)	5.66 (5.43–6.05)	0.054
Eyeball transverse diameter, mm				
ETD, Rt.	22.68 (21.92–23.32)	23.08 (22.21–23.60)	22.58 (21.83–23.28)	0.162
ETD, Lt.	22.66 (22.01–23.29)	22.80 (22.20–23.44)	22.61 (21.96–23.11)	0.286
ETD, average	22.65 (21.94–23.30)	22.98 (22.13–23.52)	22.61 (21.83–23.22)	0.204
ONSD/ETD ratio				
ONSD/ETD ratio, Rt.	0.2531 (0.2354–0.2721)	0.2407 (0.2263–0.2557)	0.2578 (0.2404–0.2768)	0.013
ONSD/ETD ratio, Lt.	0.2480 (0.2293–0.2671)	0.2377 (0.2252–0.2503)	0.2521 (0.2335–0.2723)	0.021
ONSD/ETD ratio, average	0.2506 (0.2346–0.2676)	0.2392 (0.2252–0.2532)	0.2550 (0.2386–0.2721)	0.014

Abbreviations: ETD, eyeball transverse diameter; GNO, good neurologic outcome; Lt, left; ONSD, optic nerve sheath diameter; PNO, poor neurologic outcome; Rt, right.

**Table 3 medicina-58-01233-t003:** Univariable and multivariable analysis for good neurologic outcome with variables and the ONSD/ETD ratio.

Variables	Univariable Analysis		Multivariable Analysis	
	Unadjusted OR (95% CI)	*p*-Value	Adjusted OR (95% CI)	*p*-Value
Age, year	0.928 (0.892–0.965)	<0.001	0.916 (0.840–0.997)	0.043
Sex, male	0.404 (0.146–1.119)	0.081	0.289 (0.020–4.252)	0.365
DM	0.295 (0.092–0.943)	0.040	0.881 (0.065–11.994)	0.924
Etiology, cardiac	25.667 (8.192–80.361)	<0.001	10.111 (0.423–241.721)	0.153
Etiology, respiratory	0.114 (0.025–0.518)	0.005	0.503 (0.021–12.129)	0.672
Bystander CPR	2.933 (1.063–8.097)	0.038	0.307 (0.020–4.662)	0.395
Shockable rhythm	30.600 (8.588–109.027)	<0.001	5.351 (0.349–81.932)	0.228
Pre-hospital ROSC	9.200 (3.111–27.204)	<0.001	1.569 (0.204–12.059)	0.665
No-flow time, min	0.902 (0.850–0.958)	0.001	0.924 (0.803–1.063)	0.270
Low-flow time, min	0.950 (0.905–0.998)	0.041	0.894 (0.814–0.982)	0.019
ROSC to CT interval *, min	0.987 (0.977–0.996)	0.006	0.988 (0.968–1.009)	0.259
ONSD/ETD ratio	0.000 (0.000–0.009)	0.014	0.000 (0.000–126,344,779.7)	0.173

Multivariable model fit was acceptable (Hosmer-Lemeshow; *p* = 0.420). Abbreviations: CI, confidence interval; CPR, cardiopulmonary resuscitation; CT, computed tomography; DM, diabetes mellitus; ETD, eyeball transverse diameter; ONSD, optic nerve sheath diameter; OR, odds ratio; ROSC, return of spontaneous circulation. * The interval between ROSC and brain CT.

**Table 4 medicina-58-01233-t004:** Cut-off and diagnostic value of the ONSD/ETD ratio for predicting good and poor neurologic outcomes.

	Cut-Off	Sensitivity	Specificity	PPV	NPV
ONSD/ETD ratio for predicting PNO	>0.290	0.111	1.000	1.000	0.304
ONSD/ETD ratio for predicting GNO	≤0.257	0.857	0.458	0.381	0.892

Abbreviations: ETD, eyeball transverse diameter; GNO, good neurologic outcome; NPV, negative predictive value; ONSD, optic nerve sheath diameter; PNO, poor neurologic outcome; PPV, positive predictive value.

## Data Availability

The datasets used and/or analyzed during the current study are available from the corresponding author on reasonable request.

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
