# Peer review of "The Prognostic Value of Optic Nerve Sheath Diameter/Eyeball Transverse Diameter Ratio in the Neurological Outcomes of Out-of-Hospital Cardiac Arrest Patients"

_medicina, 2022, doi:10.3390/medicina58091233_

Round 1

Reviewer 1 Report

Comments to authors:

Thank you very much for submitting manuscript related to the prognostic evaluation of patients with cardiac arrest.

I would like to hear from the authors on a few points.

#1. What are the differences from previous papers?

There are articles in the literature where the research design of the thesis appears to be very similar.

example: 

Ling, Dean-An, et al. "Optic Nerve Sheath Diameter (ONSD)/Eyeball Transverse Diameter (ETD) Ratio: An Early Indicator Associated With Poor Neurological Recovery in Cardiac Arrest Survivors." Jia-Yu and Chen, Yi-Chu and Ko, Ying-Chih and Chang, Chih-Heng and Lien, Wan-Ching and Chang, Wei-Tien and Huang, Chien-Hua, Optic Nerve Sheath Diameter (ONSD)/Eyeball Transverse Diameter (ETD) Ratio: An Early Indicator Associated With Poor Neurological Recovery in Cardiac Arrest Survivors.

#2. When using CT, the degree of image cutting (angle, number of images included) seems to be different depending on the method taken, so it is judged to be a variable that can affect the research do.

#3. Differences in age and shockable rhythm, which are known strong prognostic determinants, may act as biases. There is also a statistical difference in basic characteristics between the actual population, and if you have considered correction for this or not, I think it is appropriate to mention the limitation.

#4. Previous studies have argued that ONSD is sufficiently related to prognosis, so why calculate the ratio?

(example:

Lee, Heekyung, et al. "Predictive Utility of Changes in Optic Nerve Sheath Diameter after Cardiac Arrest for Neurologic Outcomes." International journal of environmental research and public health 18.12 (2021): 6567.

Looking at figure 2., although it is not statistically significant, it seems that there may be a sufficient difference. The p-value is close to 0.05.

Although there was some tendency in the conclusion or discussion, it is good to mention that the difference can be confirmed more clearly if the ONSD is corrected with the ETD ratio, etc.

Even if there is no significant difference in the p-value, it does not necessarily mean that there is no difference between the populations.

#5. Since there is a difference in ROSC CT time, isn't it checked at the same time?

Since there is a difference in ROSC CT time, isn't it checked at the same time? Is it possible that such a factor may have caused an error in the measurement of the optic nerve sheath?

#6. 5. It was measured by two people, but don't you provide kappa value, etc.? There is an interobserver discrepancy factor. There is no detailed explanation about whether the two people agreed at the same time or whether the min or max values were taken.

It is seemed that there is a possibility that it may act as a bias.

Best regards,

Author Response

Dear Reviewer:

We wish to express our appreciation for the assessment of our manuscript and insightful comments and suggestions.

We have made several changes in the manuscript in accordance with your suggestions.

The manuscript has benefited from these insightful suggestions. Revised areas in the manuscript are indicated in yellow highlighted text.

Our point-by-point responses to your comments are given below.

Best regards.

Reviewer 2 Report

interesting paper. Two concerns are. the small number of recruited patients and excluded ones. Moreover do you think that a brain MRI instead of CT could have added sth? And evoked potentials?  Did you also evaluate Glasgow Coma Scale at the emergency Department? Was it <8 in all patients?

Author Response

Dear Reviewer:

We wish to express our appreciation for the assessment of our manuscript and insightful comments and suggestions.

The manuscript has benefited from these insightful suggestions.

Our point-by-point responses to your comments are given below.

Best regards.

Reviewer 3 Report

Thank you very much for the opportunity of reviewing this paper about the possibile association between the optic nerve sheath diameter and the eyeball transverse  in post cardiac arrest patients. This a well written study but with rather small sample size and with some important limitations.

1) the sample size is very small. It could be probably powered enough  to compare differences between the two groups but not enough to speak about cut-off values.

2) They run a multivarible model for the probability of having a good neurological outcome. Only 28 patients had a good neurological outcome so at maximum you could have performed a bivariate analysis. If you see table 3 you can 12 covariates. I don't think this model could be reliable.

3) I would suggest to check the ROC curve. The 95%CI is too large and overcomes the line of identity. It sounds strange that the p value of the AUC could be 0.006

4) the Author did not provide us with any information about temperature control and if CT images have been obtained during or after the phase of temperature control.

5) The Author made no mention about pupillary diameter. This is an emerging prognostic value rather linked with what you wanted to address.

Author Response

(The authors gave the same response as above.)

Round 2

Reviewer 1 Report

Comments to authors:

You have been relatively faithful and faithful in explaining various issues. Thank you.

I would be very grateful if you could contribute as a good study in the future.

Best regards,

Author Response

Thank you for your comments.

We sincerely appreciate your review of our paper and your positive feedback.

Reviewer 3 Report

Thank you very much for responding to my comments. You have solved most of the problems, but I still have some concerns about the multivariable model. I thank you for the opportunity to review the literature on logistic regressions. I learned that nowadays the event-per-variable ratio is still debated (although I and others believe it is a good practice) but considering too many variables l could lead to overfitting the model. I strongly suggest inserting columns, to the left of the existing ones for multivariable analysis, to show the univariable analysis of the same variables. Then, include in the multivariable analysis at least only those with a p less than 0.1 so as to limit the number of variables to the most important ones.  With such a small number of events, I suggest that you make it clear to the reader (and the reviewer) that you have made an effort to select the variables to be included in the multivariable model. If each variable was significant in a univariable model, it would have shown the real importance of its presence in the multivariable analysis.

Author Response

Dear Reviewer:

We wish to express our appreciation for the assessment of our manuscript and insightful comments and suggestions.

We have made several changes in the manuscript in accordance with your suggestions.

The manuscript has benefited from these insightful suggestions. Revised areas in the manuscript are indicated in yellow highlighted text.

Our responses to your comments are given below.

Best regards.
